# Accurate Chromosome Identification in the Prunus Subgenus *Cerasus* (*Prunus pseudocerasus*) and its Relatives by Oligo-FISH

**DOI:** 10.3390/ijms232113213

**Published:** 2022-10-30

**Authors:** Lei Wang, Yan Feng, Yan Wang, Jing Zhang, Qing Chen, Zhenshan Liu, Congli Liu, Wen He, Hao Wang, Shaofeng Yang, Yong Zhang, Ya Luo, Haoru Tang, Xiaorong Wang

**Affiliations:** 1College of Horticulture, Sichuan Agricultural University, Chengdu 611130, China; 2Institute of Pomology and Olericulture, Sichuan Agricultural University, Chengdu 611130, China; 3Zhengzhou Fruit Research Institute, Chinese Academy of Agricultural Sciences, Zhengzhou 410100, China

**Keywords:** *Prunus* (*cerasus*), chromosome identification, oligonucleotide, FISH, rDNA

## Abstract

A precise, rapid and straightforward approach to chromosome identification is fundamental for cytogenetics studies. However, the identification of individual chromosomes was not previously possible for Chinese cherry or other Prunus species due to the small size and similar morphology of their chromosomes. To address this issue, we designed a pool of oligonucleotides distributed across specific pseudochromosome regions of Chinese cherry. This oligonucleotide pool was amplified through multiplex PCR with specific internal primers to produce probes that could recognize specific chromosomes. External primers modified with red and green fluorescence tags could produce unique signal barcoding patterns to identify each chromosome concomitantly. The same oligonucleotide pool could also discriminate all chromosomes in other Prunus species. Additionally, the 5S/45S rDNA probes and the oligo pool were applied in two sequential rounds of fluorescence in situ hybridization (FISH) localized to chromosomes and showed different distribution patterns among Prunus species. At the same time, comparative karyotype analysis revealed high conservation among *P. pseudocerasus*, *P. avium*, and *P. persica*. Together, these findings establish this oligonucleotide pool as the most effective tool for chromosome identification and the analysis of genome organization and evolution in the genus *Prunus*.

## 1. Introduction

Chinese cherry (*Prunus pseudocerasus*) belongs to the genus *Prunus* (subgenus *Cerasus*) in the Rosaceae family [1]. It originated in China as an essential native fruit tree with a distinctive rich and unique flavor, and it is cultivated widely in Southwest China [2]. *Prunus* comprises many species that are valued for their edible fruit and ornamental characteristics, and members of this subgenus are widely distributed in temperate regions. However, most cultivated species are diploid (2n = 2x = 16), with the exception of Chinese cherry (*P. pseudocerasus*, 2n = 4x = 32) and sour cherry (*P. cerasus*, 2n = 4x = 32). Several studies have demonstrated that the tetraploid *P. cerasus* was formed from natural hybridization between *P. avium* and *P. fruticosa* [3,4]. Chinese cherry is typically tetraploid; however, pentaploid and hexaploid accessions are occasionally found [5]. However, the origin of tetraploid *P. pseudocerasus* remains undetermined. Additionally, with the development of sequencing technologies, molecular markers and phylogenetic analyses have been widely applied in recent years, and there is some controversy regarding the taxonomic classification of draw cherries [6,7].

The chromosome is the critical carrier of genetic information in eukaryotic cell nuclei, and chromosome numbers, sizes, and shapes are typically stable within species [8]. The karyotype is a fundamental characteristic of the chromosome complement. Cytological studies have played an essential role in the analysis of chromosomal structural variation, evolution and phylogenetic relationships with related species [9,10]. An accurate and reliable method for chromosome identification is the foundation for karyotype analysis. However, the chromosomes in the genus *Prunus* are small in size and have similar morphology, making them difficult to distinguish; this is one of the most important reasons why there are only a few reports available on karyotypes in the genus *Prunus* [5,11,12,13]. Fluorescence in situ hybridization (FISH) was mainly dependent on the probe sequence to yield signals that could be used for chromosome identification. Repetitive DNA sequences, such as 5S/45S rDNA, telomeres, centromeres and other specific repetitive DNA, contribute a large proportion of the sequence in plant genomes and have been an essential source of FISH probes [14]. Therefore, many studies have reported the use of repetitive DNA as probes in *Cucumis* [15], *Citrus* [16], *Arachis* [17], *Ipomoea* [18], and *Triticum* [19]. In contrast, only a few studies have reported the physical mapping of rDNA in *Prunus* species, such as peach [20], almond [21], cherry rootstocks [22], and African cherry [23].

Questions about the accurate identification of chromosomes have not been thoroughly solved for most plant species. The genomes of an increasing number of species have been obtained with extensively used sequencing technology. A new approach for a FISH probe design that uses single-copy oligonucleotide sequences (40 to 50 bp) selected from a partial chromosomal region or an entire chromosome according to genome sequencing data has been assembled. The selected oligonucleotide sequence is directly synthesized and labeled as the FISH probe. This approach has been used to identify specific chromosomes successfully in cucumber [24,25,26], strawberry [27,28], rice [29], poplar [30], maize [31,32], banana [33,34], beans [35], sugarcane [36,37], oil palm [38], wheat [39,40], *Citrus* [41], *Ipomoea* [42], *Solanum* [43], and *Arachis* [17]. Approximately thirty materials spanning the genus *Prunus* have been sequenced due to their small genome size and critical economic importance as fruit trees. Remarkably, most of these genomes have been obtained by single-molecule real-time long-read sequencing or via Hi-C technology, which provide high-quality and integrated genome data [44,45,46,47,48,49,50]. Even though whole-genome sequencing has become an integral approach for detecting genome structure and genetic variation, it is unlikely to be widely conducted for most species within the near future. However, a large amount of existing genomic data is now available for the development of new FISH probes for cytogenetic analysis.

Here, we designed and synthesized an oligonucleotide pool of single-copy sequences that could be used to accurately identify all chromosomes of Chinese cherry and related species according to their FISH signal distribution pattern. We demonstrated that the single-copy sequences on the chromosomes within the genus Prunus were conserved. After two sequential rounds of FISH with the 5S/45S rDNA probes and the oligonucleotide pool, the 5S/45S rDNA distribution patterns on the chromosome exhibited a more significant variation. We also performed a comparative karyotype analysis of the closely related species *P. pseudocerasus* and *P. avium*, and their distantly related species *P. persica*, which exhibited high similarity. 

## 2. Results

### 2.1. Phylogenetic Relationships Established Based on 45S rDNA

We assembled the completed 45S rDNA sequences using a low-coverage genome sequence. In all taxa of *Prunus*, the sequence length of 45 S rDNA varied from 5774 to 5793 bp. Sequence alignments showed that the 18 S, 5.8 S, and 28 S rDNA subunits had the same lengths among different taxa (1808 bp, 159 bp, and 3347 bp, respectively) and that their sequences are highly conserved, with only a few mutated bases. In contrast, ITS1 ranged in length from 228 to 247, ITS2 ranged from 229 to 233, and the sequence variation in these regions was more apparent (Figure 1, Appendix A). 

Subsequently, the complete 45S rDNA sequences were used to construct the phylogenetic tree with the maximum-likelihood method (Figure 2). The phylogenetic tree showed that the species in the *Prunus* genus were mainly divided into two clades. All species in the subgenus *Cerasus* were grouped into one clade. The two species of dwarf cherry (*P. humilis* and *P. tomentosa*) grouped with *P. armeniaca*, *P. salicina*, *P. dulcis* and *P. persica*, showing that the dwarf cherry is more distantly related to the subgenus *Cerasus*.

### 2.2. Development of Oligo-FISH Probes Using Single-Copy Sequences from Chinese Cherry

We designed 14 sublibraries distributed across specific regions on eight chromosomes of Chinese cherry according to the repeat sequences and oligonucleotide distribution (Figure 3, Table 1). Each sublibrary contained a total of 5146 (Contig_20 on Chr_8) to 7960 (Contig_01 on Chr_5) oligos and spanned a genomic region ranging from 1.40 Mb (Contig_25 on Chr_2) to 2.14 Mb (Contig_29 on Chr_1). Moreover, most sublibraries achieved a density of more than 3 oligos per kb except for Contig_20 (Chr_8, 2.71 oligos/kb), which contains more repetitive sequences.

Subsequently, oligos from each sublibrary on the same chromosome were flanked with the same internal primers (P1 to P8 for Chr_1 to Chr_8) to allow amplification by specific internal primers (Table 1, Appendix A), thus allowing accurate identification of unique chromosomes. Additionally, one external primer pair (W1 or W2) was added to all oligos to allow the use of green and red fluorescence modifier primers for amplification by PCR (Figure 3, Table 1). Thus, six regions of green and eight regions of red signals were generated which could uniquely identify each chromosome for *P. pseudocerasus* simultaneously. Therefore, each synthetic oligo contained a 50 bp genomic sequence and two pairs of external and internal primers (Appendix A). All oligonucleotides from the 14 sublibraries were synthesized in a single pool.

We also confirmed that the selected oligonucleotide sequences from Chinese cherry could align with the pseudochromosomes of closely related species. Their distribution showed that these 14 sublibraries tended to cluster into specific chromosomal locations with similar barcode distributions to Chinese cherry except for Chr_1 of *P.* × *yedoensis* (Appendix A), which could be due to the lower completeness of the assembled genome. Thus, we were able to use this oligo pool to accurately identify the target chromosomes in other species in *Prunus*.

### 2.3. Chromosome Identification in Chinese Cherry and Related Species

Repeat sequences are typically a critical source of probes for molecular cytogenetics. Here, three types of repetitive oligo probes (Oligos_102, Oligos_149, and Oligos_159, Appendix A) were prepared for FISH. Unexpectedly, their signals were all distributed in the telomeric regions of chromosomes without apparent features other than interspecies variation (Appendix A). We also carried out 5S/45S rDNA probes to identify their location on the chromosome (Appendix A). This showed that one (two) pair of chromosomes bore 5S rDNA in the diploid (tetraploid) species *P. avium*, *P. tomentosa*, *P. humilis*, and *P. salicina* (*P. cerasus*), which means the 5S rDNA could be used to identify the specific chromosome in those species. Still, it could not distinguish the chromosome in other diploid (tetraploid) species because they have two (four) pairs of chromosomes with 5S rDNA probe signals. However, the 45S rDNA probes were not available for directly identifying the specific chromosome because they presented two or three (six) pairs of chromosomes with signals in the diploids (tetraploid) species. Coincidently, one (two) pair of chromosomes bore 5S and 45S rDNA probes simultaneously in the diploid (tetraploid) species of *P. yedoensis* (P. *pseudocerasus*).

In this study, probes with repetitive sequences could not be used for each chromosome identification. Thus, we used the oligo pool strategy to amplify two sets of sublibraries as probes with external primers (W1 was labeled with FAM and W2 was labeled with TAMRA) to identify all chromosomes in the same metaphase cells from Chinese cherry. The eight green and six red conserved FISH signals derived from the probes of these two sublibraries in the tetraploid and pentaploid species *P. pseudocerasus* (Figure 4) matched the predicted patterns (Figure 3). The signals formed a clear barcode that could accurately identify the eight chromosomes from Chinese cherry. We then performed FISH on the other ten *Prunus* species and observed signal patterns similar to those seen in the Chinese cherry (Figure 5, Appendix A). Thus, probes from the two sublibraries could be used to identify all chromosomes from *Prunus*.

### 2.4. Distribution of 5S rDNA and 45S rDNA in Different Species

To map the 5S/45S rDNA to the chromosomes, after the first round of FISH using the two sets of oligo sublibrary probes (Figure 6(a1–e1)), the slide was washed and reprobed with 5S/45S rDNA probes (Figure 6(a2–e2)). These 14 sublibraries produced oligo-based barcode signals (Figure 3 and Figure 4). In tetraploid *P. pseudocerasus*, the 5S rDNA is located on the proximal regions of the short arms of Chr_7 and Chr_8. The 45S rDNA is located in the terminal regions of the short arms of Chr_4, Chr_5, and Chr_7. In addition, the lack of three 45S rDNA signals was observed on Chr_4 in the pentaploid accession of *P. pseudocerasus* (Figure 6(e1,e2)). This might indicate that the signal is too weak to be detected.

Subsequently, we further confirmed the locations of the 5S/45S rDNA on the chromosomes of other species of *Prunus* (Appendix A). Only *P. yedoensis* presented a 5S/45S rDNA distribution pattern similar to that of Chinese cherry (Appendix A). Other species showed distinct distribution patterns, particularly for 5S rDNA, which showed relatively large variations among different species, even closely related diploid species (*P. dulcis* and *P. persica*; *P. salicina* and *P. armeniaca*; *P. campanulata*, *P. yedoensis*, and *P. avium*), without apparent associations between species phylogeny and 5S rDNA distribution (Figure 7, Appendix A). The 5S rDNA signal was identified in the centromeric regions of Chr_6 and the proximal region of Chr_8 (*P. campanulata*, *P. armeniaca*, and *P. persica*), the centromeric regions of Chr_3 and Chr_6 (*P. dulcis*), or the centromeric regions of Chr_6 (*P. humilis*, *P. tomentosa*, and *P. salicina*), or the proximal region of Chr_8 (*P. avium* and *P. cerasus*). In contrast, the 45S rDNA distribution pattern was conserved. Additionally, its distribution showed a close relationship with phylogeny (Figure 7, Appendix A). It was detected in the terminal regions of the short arms of Chr_4 and Chr_7 for all species, and it was also detected on the terminal regions of Chr_5 in the subgenus *Cerasus* (*P. pseudocerasus*, *P. campanulata*, *P. avium*, *P. cerasus*, and *P. yedoensis*), the centromeric regions of Chr_5 for the closely related species *P. dulcis* and *P. persica*, or the centromeric regions of Chr_3 for the closely related species *P. salicina* and *P. armeniaca*.

Moreover, a faint signal was occasionally detected on Chr_8 for some materials, leading to confusion with Chr_4. To avoid such potential ambiguity, we used the specific chromosome oligonucleotide probes from Chr_4 or Chr_8 amplified with internal primers to further identify the chromosome. Among all analyzed species, we observed that the Chr_4 contained the 45S rDNA signal, and the Chr_8 contained the 5S rDNA signal except for *P. humilis*, *P. tomentosa*, *P. salicina*, and *P. dulcis* (Appendix A). This combination presented clear signals that could accurately identify the chromosome.

### 2.5. Comparative Karyotype Analyses

To reveal the karyotype stability among the Prunus species, we performed a comparative analysis among *P. pseudocerasus*, *P. avium*, and *P. persica*. *P. pseudocerasus* displayed closer relationships with *P. avium* and was more distantly related to *P. persica* among the species we used in this study. After identifying all chromosomes in the same metaphase cells according to the barcoding patterns, the chromosomes of each species were measured in five complete metaphases. Their karyotype ideograms are shown in Figure 8. Comparative analysis revealed that the karyotypes in these three species were highly conserved (Figure 8, Appendix A).

## 3. Discussion

Accurate chromosome identification is a prerequisite and basis for cytogenetics research. Generally, the chromosomes in most *Prunus* species are very small (lengths ranging from 0.84 to 2.81 μm) and show a similar morphology [13]. In earlier studies, combined cytogenetic techniques (C-banding, BAC-FISH and 5S/45S rDNA-FISH) were often applied to horticultural crops such as *Citrus* [51,52], *Fragaria* [53], *Cucumis* [54,55], and *Solanum* [56,57] for a karyotype analysis, to identify chromosomal structural variants, and to investigate genome evolution, but research in *Prunus* has been limited to the characterization of karyotypes and 5S/45S rDNA in only a few species. In this study, we performed a 5S/45S rDNA analysis of 11 species of *Prunus*. Several different signal distribution patterns were observed: two pairs of 45S rDNA signals and one pair of 5S rDNA signals in the diploid species *P. humilis* and *P. tomentosa*; three pairs of 45S rDNA signals and one pair of 5S rDNA signals in the diploid species *P. avium* and *P. salicina*; three pairs of 45S rDNA signals and two pairs of 5S rDNA signals in the diploid species *P. campanulata*, *P. yedoensis*, *P. armeniaca*, *P. dulcis*, and *P. persica*; and six pairs of 45S rDNA signals and four (two) pairs of 5S rDNA signals in tetraploid *P. pseudocerasus* (*P. cerasus*). 

Among all of them, one (two) pair of 5S rDNA signals and one (two) pair of 45S rDNA signals were located on the same chromosomes in *P. yedoensis* (*P. pseudocerasus*) (Appendix A). In parallel, there are two (four) chromosomes with longer lengths in the diploid (tetraploid), while the other chromosomes all have similar lengths. Consequently, it is impossible to identify individual chromosomes. Repetitive DNA sequences, as a vital source of FISH probes, have been broadly applied in many plants [14], especially for peanut [17,58] and wheat [59,60]. However, the three probes that we identified previously could not be used for chromosome identification due to their locations at the terminus of chromosomes, which was due to the lower content or variety of tandem repeat sequences in their genomes [61]. Therefore, we used a combination of genomic data and oligo pool synthesis techniques to design a chromosome-specific single-copy oligonucleotide pool that could be used to accurately identify chromosomes. Similar methods were previously reported for use in cucumber [26], oil palm [38], maize [32], beans [35], potato, tomato, and eggplant [62], rice [63], and Japanese morning glory [42]. This will further improve the accuracy of karyotype analysis and chromosomal structural variant detection within species. This method also has two distinct strengths relative to previous techniques for cytogenetic karyotyping [62,64]. First, all chromosomes can be easily, unambiguously, and simultaneously identified in the mid-phase of mitosis. Second, the chromosome nomenclature is assigned based on the peach genome assembly, which is convenient for comparative cytogenetic analysis among closely related species.

Previous studies have suggested that the pseudochromosomes in the genomes of sweet cherry, plum, apricot, mei, almond and peach have a high level of overall synteny, albeit with minor interchromosomal rearrangements [46,48,49], and that the divergence time between sweet cherry and peach was approximately 10 million years ago (MYA) [45,65]. In this study, the oligo pool covered a relatively small proportion of the entire genome (23.17 Mb). It showed similar barcode signals among *Prunus* species without discernible variations in chromosome structure or signal intensity. Additionally, the comparative karyotype analysis of two closely related species (Chinese cherry and sweet cherry) and a distantly related species (peach) showed that the chromosome structures were highly conserved. Thus, all these results indicated the high conservation of the genome, as expected, among the *Prunus* species. Previous reports have confirmed a remarkably conserved karyotype in species that diverged from a common ancestor in the genera *Fragaria* (6.4 MYA) [28], *Solanum* (5–8 MYA) [62], *Citrus* (9 MYA) [41], *Cucumis* (12 MYA) [24], and *Populus* (14 MYA) [30] and is consistent with our results. Furthermore, in this study, we identified eight different types of 5S/45S rDNA distribution patterns, which illustrated that the 5S/45S rDNA in Prunus is in a highly dynamic state. The distribution pattern of 5S/45S rDNA signals on the chromosomes differed mainly due to 5S/45S rDNA copy number deletion/amplification, 5S/45S rDNA-containing fragment inversions/translocations, or 5S/45S rDNA unequal crossing over [66,67,68]. Similar results were also reported in *Arachis* [17], *Cucumis* [69], and *Citrus* [41]. 

Polyploidy is especially common in plants. Chinese cherry and sour cherry are the main two polyploid cultivated species in the genus *Prunus*, which are commonly tetraploid, while pentaploid or hexaploid individuals have also been detected in Chinese cherry [13]. Polyploidy can emerge directly through crossing within a single species (autopolyploidy) or through interspecific hybridization (allopolyploidy) [70]. Comparative karyotyping analysis among species has been used to infer their phylogenetic relationships or the origin of polyploid species [71]. However, karyotyping in this study showed high similarity even between distantly related species between Chinese cherry and peach due to their short divergence time (Figure 8). Thus, the comparative karyotyping analysis has little significance. Species that are more closely related have more similar 5S/45S rDNA distribution patterns [28,69]. Promisingly, *Prunus* species showed many different 5S/45S rDNA distribution patterns. This can provide valuable insights into the origin and evolution of polyploid species. In tetraploid *P. cerasus*, the two pairs of 5S rDNA signals had different signal intensities on the homologous chromosome of Chr_8, and its distribution pattern was also similar to that of *P. avium*, which partially coincides with the inference that *P. cerasus* was formed from hybridization between the species of *P. avium* and *P. fruticosa* in previous reports [3,4]. Regrettably, no *P. fruticosa* material was collected in the present study, and FISH could not be conducted for further confirmation. In addition, the origin of tetraploidy in *P. pseudocerasus* has remained undetermined. In terms of the 5S/45S rDNA distribution patterns, the tetraploid *P. pseudocerasus* was similar to diploid *P. yedoensis* and differed from *P. campanulata* and *P. avium*. Simultaneously, 5S/45S rDNA distribution patterns were relatively stable on the tetraploid and pentaploid regions of *P. pseudocerasus*, except for the lack of three 45S rDNA signals on Chr_4 that was observed in its pentaploid accession of *P. pseudocerasus*.. These results imply that *P. pseudocerasus* may have been formed from autotetraploidy of closely related species to *P. yedoensis* or allotetraploidy by interspecific hybridization between two diploid species that are closely related to *P. yedoensis*.

## 4. Materials and Methods

### 4.1. Sequencing Reads and Phylogenetic Analyses

Paired-ended sequencing reads (2 × 90 bp) were obtained from 11 true cherries (eight *P. pseudocerasus*, two *P. avium*, and one *P. yedoensis*) and two dwarf cherries (one each of *P. humilis* and *P. tomentosa*) that were sequenced with Illumina HiSeq 2000 (BGI, Shenzhen, China) by our research group (Appendix A). Additionally, the raw sequencing reads from two true cherries (*P. campanulata* and *P. cerasus*), four closely related species (*P. armeniaca*, *P. salicina*, *P. dulcis*, and *P. persica*) and one more distantly related species (*Fragaria vesca*) were downloaded from the SRA (Sequence Reads Archive, https://www.ncbi.nlm.nih.gov/sra/, accessed on 12 November 2019). Detailed information is listed in Appendix A. 

The 45S rDNA sequence was assembled using the GetOrganelle pipeline (version = 1.7.5.0, k-mer length 21, 55 and 79) with a low-coverage genome sequence from 20 taxa, and sequences with a coverage depth greater than 100 were retained [72]. Multiple 45S rDNA sequences were aligned with default parameters using MAFFT (version = 7.489), with the sequence of *Kageneckia angustifolia* (GI: 1824670479) used as the outgroup. The phylogenetic tree was reconstructed using the maximum-likelihood method with the best-fit model and 1000 bootstrap replicates in MEGAX.

### 4.2. Plant Materials and Chromosome Preparation

For FISH mapping, the seeds of *P. pseudocerasus* and related species were used for chromosome preparations (Table 2). These included five taxa with significantly different phenotypes in Chinese cherry collected from Sichuan Province and Yunnan Province and maintained in Chongzhou city (Sichuan Province). Root tips were harvested from germinated seeds at 25 °C and pretreated with 2 mM 8-hydroxyquinoline for 4 h at 4 °C. Then, the cells were fixed in Carnoy’s fixative for 24 h at 4 °C. Chromosomes were prepared via the slide-drop method, as described in published protocols [41]. Finally, at least three slides with more than three well-spread mitotic metaphase chromosomes were used for the FISH experiments.

### 4.3. Chromosome-Specific Oligonucleotide Probe Development

The oligonucleotide probes were designed using Chorus software [64], following previous reports with minor modifications [26]. Briefly, the repetitive sequences from each contig of Chinese cherry were filtered using RepeatMasker (http://www.repeatmasker.org/, accessed on 8 January 2020), and the remaining sequences were then divided into oligos (50 nt) in a step size of 5 nt. Each oligo was aligned to the Chinese cherry draft genomes (unpublished) in order to filter out oligos with more than 75% similarity to other 50 nt sequences in the genome. It has been reported that a density of more than 1.5 oligos per kb can produce clear FISH signals on meiotic chromosomes [24]. Here, we adopted more than two oligos per kb to ensure clear signals. Subsequently, the selected oligonucleotide sequences were also aligned to the pseudochromosomes of *P. avium* [46], *P.* × *yedoensis* [65], *P. salicina* [48], *P. armeniaca* [49], *P. dulcis* [50], and *P. persica* [73] with Magic-BLAST (version = 1.5, default parameters) to investigate the homology of the oligonucleotide derived from *P. pseudocerasus* to other *Prunus* species. The genome assembly information is listed in Appendix A.

### 4.4. Repetitive and Single-Copy Oligo Probe Synthesis

Three oligo probes were designed from species-specific satellite sequences (monomer lengths 102 bp, 149 bp, and 159 bp) that we identified previously (Appendix A) [61]. Additionally, the highly conserved 45S rDNA sequences were assembled in this study, and 5S rDNA sequences obtained from the 5S rDNA database (https://combio.pl/rrna, accessed on 3 October 2021) were also used to design probes with their highly conserved regions using FISH. Two prelabeled oligonucleotide probes spanned the 5S rDNA, and four prelabeled oligonucleotide probes spanned the 18S rDNA (Appendix A). These oligo probes were synthesized by Generay (Shanghai, China).

The oligonucleotide sequences that were selected on different chromosomes were flanked with specific internal primers (Table 1, Appendix A). Each flanking oligo also contained an external primer pair that could be used to simultaneously identify all chromosomes with a barcode-like signal, as reported previously [26]. The oligo pool was synthesized by Twist Biosciences (Twist, https://www.twistbioscience.com/, accessed on 1 March 2020). Primers containing 5’FAM or 5’TAMRA were synthesized by LoGenBio (Shanghai, China). 

### 4.5. FISH and Karyotyping

The oligo probe derived from the repeated sequence was synthesized and prelabeled with FAM or TAMRA and thus could be directly used as a probe. The oligo pool of single-copy oligonucleotide sequences could be amplified using the PCR protocol with their external or internal primers to generate probes for FISH that can produce barcode signals or chromosome-specific signals [26]. The FISH procedure was conducted as described previously with slight modifications [17,26]. Briefly, 20 µL of hybridization mixture solution contained 50% (*v*/*v*) deionized formamide, 10% (*w*/*v*) dextran sulfate, 2 × SSC, and 50–80 ng of prelabeled oligo probes (100–200 ng for the single-copy oligo pool). The spread chromosomes were dehydrated in 100% ethanol, and then the probe mixture was added to the chromosome slide, covered with a coverslip, denatured for 5–8 min at 85 °C, and incubated at 37 °C (3–4 h for prelabeled oligo probes; 10–12 h for the nonrepetitive oligo library). Subsequently, the hybridized chromosomes were washed twice with 2 × SSC for 5 min at 37 °C and then for an additional 5 min at room temperature. Chromosomes were counterstained in 2 ng µL-1 DAPI (4’,6-diamidino-2-phenylindole) in Vectashield antifade medium (Solarbio, Beijing, China) after the slides were dried.

For sequential rounds of FISH, slides with high-quality metaphase chromosomes were retained after the first round of FISH and then washed three times with 1 × PBS for 10 min to remove the probes. Afterward, they were denatured again in 70% formamide at 75 °C for 3 min. The slides were sequentially dehydrated in 70%, 85%, and 100% ethanol at room temperature for 5 min and then rehybridized with different probes.

Images were captured with a DP-70 CCD camera attached to a BX53 fluorescence microscope (Olympus, Tokyo, Japan) using CellSens Standard Software. Image-Pro Plus software (Version = 6.0) was used to adjust the brightness and contrast and to measure the short and long arms in five complete metaphase cells to calculate the relative length of chromosomes. 

## Figures and Tables

**Figure 1 ijms-23-13213-f001:**
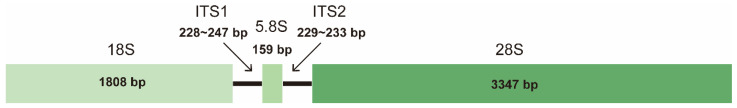
The structure between the 45S rDNA of *P. pseudocerasus* and its relatives.

**Figure 2 ijms-23-13213-f002:**
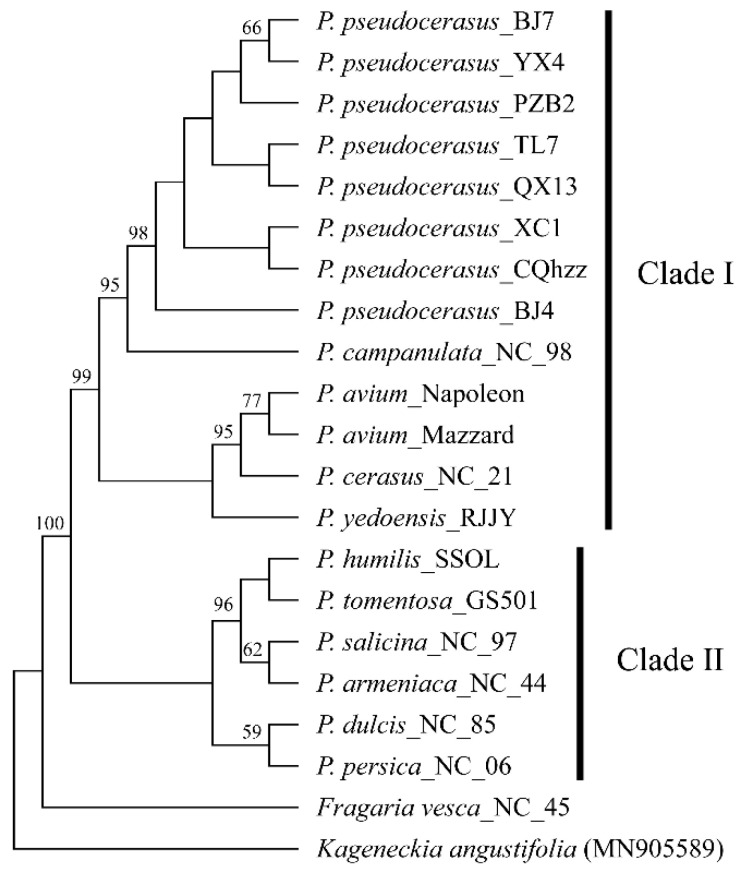
A phylogenetic tree of the genus *Prunus* was constructed based on complete 45S rDNA sequences. The ML phylogenetic tree was reconstructed using MEGAX with the “Tamura-Nei” model. Numbers at nodes are bootstrap values (as percentages) from 1000 replicates. Only values higher than 50% are shown. The 45S rDNA from the species *Kageneckia angustifolia* was used as an outgroup.

**Figure 3 ijms-23-13213-f003:**
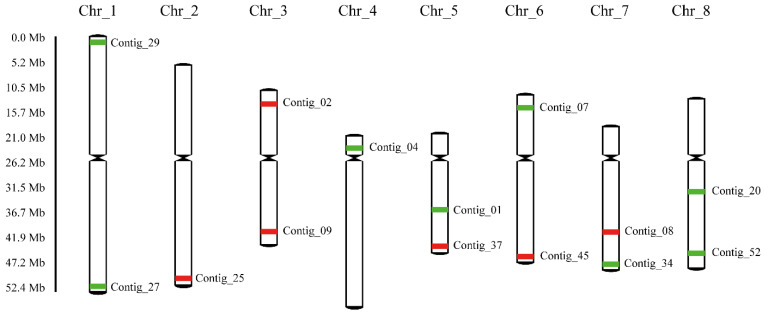
A model ideogram showing the locations of the single-copy oligo pool that produced the signals on the chromosomes of the Chinese cherry. Oligos were selected from 14 regions distributed on the pseudochromosome and colored red/green. The presence of six red and eight green regions showed that two different flanking external primer pairs could be amplified with their primers. The eight chromosomes could be distinguished based on the distribution pattern, i.e., the number and location of the red/green signals. Chromosome nomenclature followed pseudochromosomes in the draft genome of peach. The centromere positions on the eight chromosomes were determined based on the locations of centromere repeat sequences.

**Figure 4 ijms-23-13213-f004:**
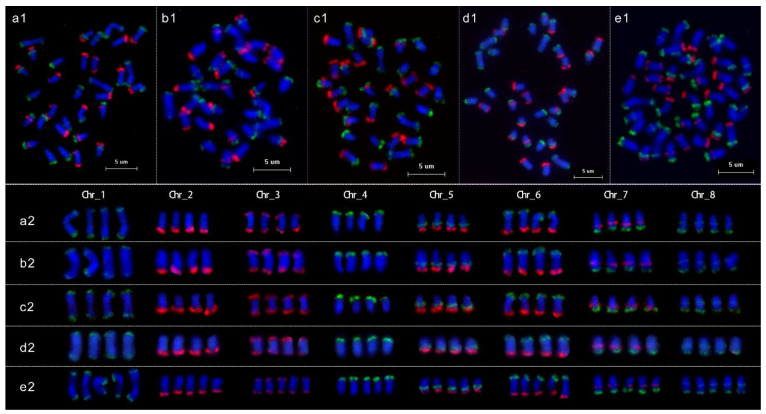
Accurate chromosome identification in metaphase spreads from Chinese cherry. (**a1**–**e1**) FISH mapping of the two single copy oligonucleotide probe sublibraries on chromosomes. *P. pseudocerasus*_HC (**a1**), *P. pseudocerasus*_BZ (**b1**), *P. pseudocerasus*_HF (**c1**), *P. pseudocerasus*_PJHH (**d1**), and *P. pseudocerasus*_FM2 (**e1**), respectively. (**a2**–**e2**) Homologous chromosomes were digitally excised from (**a1**–**e1**) and paired.

**Figure 5 ijms-23-13213-f005:**
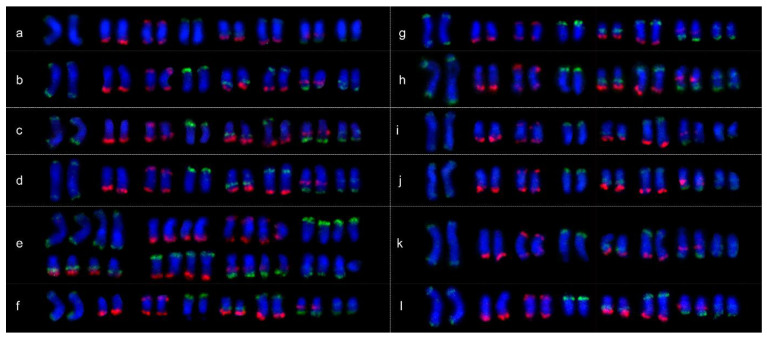
Chromosome identification in metaphase spreads of species closely related to Chinese cherry. (**a**) *P. avium*_Mazzard, (**b**) *P. avium*_Van, (**c**) *P. yedoensis*_DJYH, (**d**) *P. campanulata*_ZHYT, (**e**) *P. cerasus*_SYT, (**f**) *P. humilis*_Ouli, (**g**) *P. tomentosa*_red_fruit, (**h**) *P. tomentosa*_white_fruit, (**i**) *P. salicina*_SYL, (**j**) *P. armeniaca*_DGX, (**k**) *P. dulcis*_BT, and (**l**) *P. persica*_MT.

**Figure 6 ijms-23-13213-f006:**
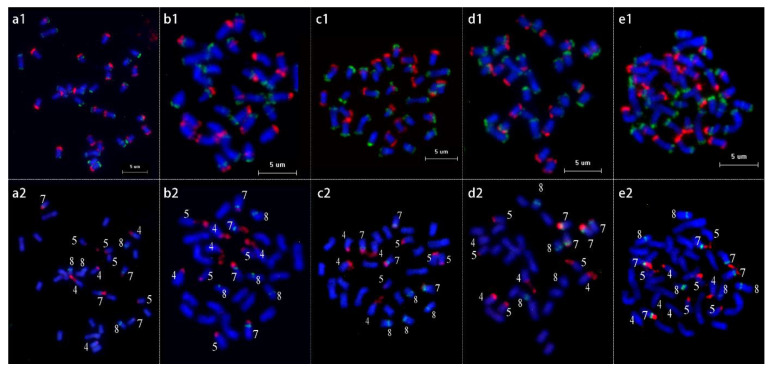
FISH of 5S rDNA and 45S rDNA accurately identified five Chinese cherries. (**a1**–**e1**) First round of FISH using two sets of oligo sublibrary probes on metaphase chromosomes for chromosome identification. *P. pseudocerasus*_HC (**a1**), *P. pseudocerasus*_BZ (**b1**), *P. pseudocerasus*_HF (**c1**), *P. pseudocerasus*_PJHH (**d1**), and *P. pseudocerasus*_FM2 (**e1**). (**a2**–**e2**) The same cells in (**a1**–**e1**) reprobed with using 5S rDNA (green) and 45S rDNA (red) probes.

**Figure 7 ijms-23-13213-f007:**
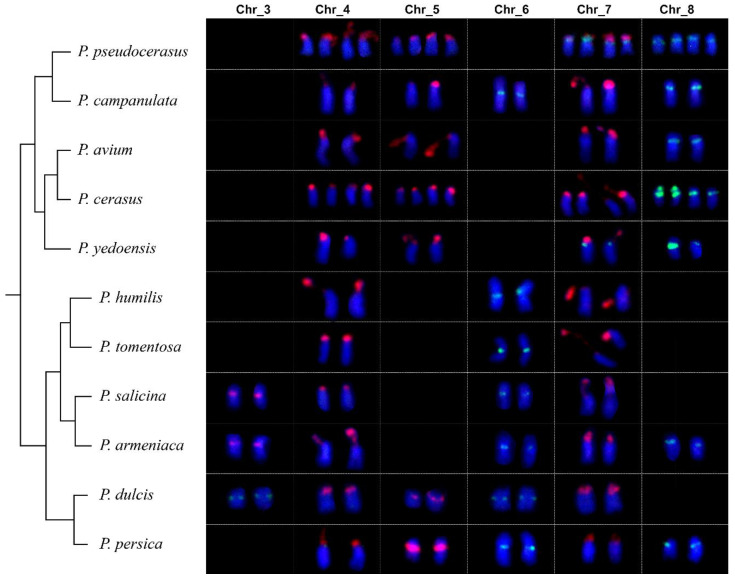
Variation in the distribution of 5S rDNA (green) and 45S rDNA (red) among *Prunus*. The phylogeny of the eleven species in this study was constructed based on 45S rDNA.

**Figure 8 ijms-23-13213-f008:**
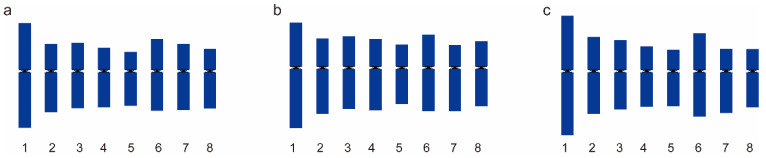
Ideogram of the karyotypes of *P. pseudocerasus*_HC (**a**), *P. avium*_Mazzard (**b**), and *P. persica*_MT (**c**). Chromosome nomenclature according to the genome *P. persica*.

**Table 1 ijms-23-13213-t001:** Numbers of oligos and contig positions of 14 regions selected for oligo-FISH probe development.

Contigs	Contigs Length (bp)	Oligo Number	Corresponding Chromosome	Region Length (bp)	Density Oligos/kb	Primers Flanking Oligos *
Contig_27	2,252,344	6790	Chr_1	1,699,807	3.99	W1.F + P1.F—P1.R + W1.R
Contig_29	2,199,550	7513	Chr_1	2,141,321	3.42	W1.F + P1.F—P1.R + W1.R
Contig_25	2,290,169	7413	Chr_2	1,399,377	5.30	W2.F + P2.F—P2.R + W2.R
Contig_02	5,439,059	6568	Chr_3	1,594,437	4.11	W2.F + P3.F—P3.R + W2.R
Contig_09	4,505,147	6618	Chr_3	1,462,969	4.41	W2.F + P3.F—P3.R + W2.R
Contig_04	5,080,108	6964	Chr_4	1,480,858	4.64	W1.F + P4.F—P4.R + W1.R
Contig_01	7,149,121	7960	Chr_5	1,998,387	3.98	W1.F + P5.F—P5.R + W1.R
Contig_37	1,734,690	7162	Chr_5	1,676,082	4.21	W2.F + P5.F—P5.R + W2.R
Contig_45	1,603,937	7120	Chr_6	1,541,098	4.45	W2.F + P6.F—P6.R + W2.R
Contig_07	4,762,281	7080	Chr_6	1,765,125	3.93	W1.F + P6.F—P6.R + W1.R
Contig_34	2,027,033	6780	Chr_7	1,516,158	4.52	W1.F + P7.F—P7.R + W1.R
Contig_08	4,598,159	5407	Chr_7	1,598,107	3.38	W2.F + P7.F—P7.R + W2.R
Contig_20	2,726,937	5146	Chr_8	1,879,060	2.71	W1.F + P8.F—P8.R + W1.R
Contig_52	1,520,798	6773	Chr_8	1,418,134	4.84	W1.F + P8.F—P8.R + W1.R

* W1.F-W2.F and W1.R-W2.R represent external forward and reverse primers flanking oligos from specific segments, while P1.F-P8.F and P1.R-P8.R represent internal forward and reverse primers (Appendix A).

**Table 2 ijms-23-13213-t002:** Plant materials used in this study for oligo-FISH on mitotic metaphase chromosomes.

Species	Accession Name or Code	Ploidy	Origin (Location)
True cherry			
*Prunus pseudocerasus*	HC; BZ; HF; PJHH; FM2 *	2n = 4x = 32	Chongzhou City, Sichuan Province, China
*Prunus yedoensis*	DJYH	2n = 2x = 16	Chengdu City, Sichuan Province, China
*Prunus campanulata*	ZHYT	2n = 2x = 16	
*Prunus avium*	Mazzard; Van	2n = 2x = 16	Zhengzhou Fruit Institute, Chinese Academy of Agriculture Sciences
*Prunus cerasus*	SYT	2n = 4x = 32	
Dwarf cherry			
*Prunus humilis*	Ouli	2n = 2x = 16	Zhengzhou Fruit Institute, Chinese Academy of Agriculture Sciences
*Prunus tomentosa*	red_fruit; white_fruit	2n = 2x = 16	
Closely related species			
*Prunus armeniaca*	DGX	2n = 2x = 16	Akesu, Xinjiang Uygur Autonomous Region, China
*Prunus dulcis*	BT	2n = 2x = 16	Luntai, Xinjiang Uygur Autonomous Region, China
*Prunus salicina*	SYL	2n = 2x = 16	Chengdu City, Sichuan Province, China
*Prunus persica*	MT	2n = 2x = 16	

* represents that this accession is a pentaploid (2n = 5x = 40).

## Data Availability

Not applicable.

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
