# Peer review of "Accurate Chromosome Identification in the Prunus Subgenus *Cerasus* (*Prunus pseudocerasus*) and its Relatives by Oligo-FISH"

_ijms, 2022, doi:10.3390/ijms232113213_

Round 1

Reviewer 1 Report

The reviewed paper presents original and novel data that will be of interest for the scientific community researching the genus Prunus as well as a broader audience. The overall quality of the manuscript is good and the methodology is accurate. I recommend accepting the manuscript after minor revisions according to the remarks listed below.

-        For a better read, I suggest moving the paragraph describing the number of 5S/45S rDNA signals (lines 149 – 160) from subsection 2.3 to subsection 2.4, which is specifically dedicated to the description of 5S/45S rDNA distribution in the investigated species.

-        In the Introduction the authors mention the occurrence of only tetraploid P. cerasus and P. pseudocerasus. However, later in the manuscript they describe the results also for pentaploid (introduced in line 151) and hexaploid (introduced in line 196) accessions of P. pseudocerasus. The authors should better explain the variation in the ploidy level in this species either in the Introduction or the Results. It would be also helpful to add information about the ploidy level of each species and accession used in the study in Table 2.

-        In Figure 6 e_1 and 6 e_2 I counted only 40 chromosomes, which would indicate that this is a pentaploid, not a hexaploid accession.

-        When referring to P. pseudocerasus with a particular ploidy level, it would be appropriate to use the term “accession.” For example: instead of “was observed on Chr_4 in the hexaploid species” (line 196) it should be “was observed on Chr_4 in the hexaploid accession of P. pseudocerasus.” The same comment applies to line 318.

-        Lines 316-317: “the tetraploid, pentaploid and hexaploidy regions of P. pseudocerasus” should be replaced with “the tetraploid, pentaploid and hexaploid accessions of P. pseudocerasus.”

-        Lines 195 – 197: The authors should be careful in using the term “loss of three 45S rDNA signals,” particularly if they indicate in the next sentence that the signals (and by extension the probe target sequences) might be present, only too weak to be detected.  The term “loss” is usually used in the context of sequence elimination. The words “lack of three 45S rDNA signals” would be more appropriate.

-        Lines 217-221: The authors did not state explicitly what was the result of the experiment. A summarizing sentence should be added.

-        Lines 235-236: I suggest specifying that the karyotypes are stable (the term “conserved” would be better) with regard to their morphometric features. The 5S/45S rDNA polymorphism observed by the authors indicates that there might be a certain degree of instability in terms of internal chromosome structure.

Reviewer 2 Report

Dear Authors,

     I have reviewed this manuscript with pleasure and I think it is worth to be publish although some corrections should be done.

Specific comments and questions to be addressed:

In Plant Materials and chromosome preparation chapter there is no information about the numbers of slides which were assessed for each of analysed genotype (plant species).

Please add some more precise information.

In the Table 2, titled: Plant materials used in this study for molecular cytogenetics,  in general 11 species were indicated as analysed but in the text in lines 323-325, the Authors mentioned about 11 true species and two additional dwarf species. Please define precisely and rewrite this chapter.

In Results description in 2.3. Chromosome identification in Chinese cherry and related species chapter in my opinion for identification the particular chromosomes in Pronus species, there should be numbers of signals counted for particular chromosomes precisely indicated , maybe even in the separate table for each analysed genotype. Although the Authors mentioned that

„The 5S rDNA probes generated eight (ten) signals in the tetraploid (pentaploid) species of P. pseudocerasus and four signals in the tetraploid species P. cerasus. At the same time, the 45S rDNA probes generated 12 signals in the tetraploid (pentaploid) species P. pseudocerasus and tetraploid species P. cerasus (Supplementary Figure S3 a_1-a_5, e). For other diploid species, the 5S rDNA presented two (P. avium, P. tomentosa, P. humilis, and P. salicina) or four (P. yedoensis, P. campanulata, P. armeniaca, P. dulcis, and P. persica) signals, and 45S rDNA presented four (P. tomentosa, and P. humilis) or six (P. avium, P. yedoensis, P. campanulata, P. salicina, P. armeniaca, P. dulcis, and P. persica) signals. The 5S rDNA and 45S rDNA were present on the same chromosome with four (two) signals in the species P. pseudocerasus (Prunus yedoensis) (Supplementary Figure S3 a_1-a_5, c).”

it seems insufficient for comparison and identification of the particular chromosomes in analysed plant species. It should be easier for the readers if the Authors could prepare the separate table.

Best regards,
